# Effectiveness of exercise intervention during pregnancy on high-risk women for gestational diabetes mellitus prevention: A meta-analysis of published RCTs

Georgios I. Tsironikos[1], Konstantinos Perivoliotis[2], Alexandra Bargiota[3], Elias Zintzaras[4], Chrysoula Doxani[4], Athina Tatsioni[5]*

1 Department of Medicine, University of Thessaly, Larissa, Greece, 2 Department of Surgery, University Hospital of Larissa, Larissa, Greece, 3 Department of Internal Medicine-Endocrinology, University Hospital of Larissa, Larissa, Greece, 4 Department of Biomathematics, University of Thessaly, Larissa, Greece, 5 Department of Research Unit for General Medicine and Primary Health Care, University of Ioannina, Ioannina, Greece

* atatsion@uoi.gr

**Data Availability Statement:** All relevant data are within the paper and its Supporting Information files.

## Abstract

### Objective

We aimed at investigating the preventive role of exercise intervention during pregnancy, in high-risk women for gestational diabetes mellitus (GDM).

### Materials and methods

We searched PubMed, CENTRAL, and Scopus for randomized controlled trials (RCTs) that evaluated exercise interventions during pregnancy on women at high risk for GDM. Data were combined with random effects models. Between study heterogeneity (Cochran's Q statistic) and the extent of study effects variability [$I^2$ with 95% confidence interval (CI)] were estimated. Sensitivity analyses examined the effect of population, intervention, and study characteristics. We also evaluated the potential for publication bias.

### Results

Among the 1,508 high-risk women who were analyzed in 9 RCTs, 374 (24.8%) [160 (21.4%) in intervention, and 214 (28.1%) in control group] developed GDM. Women who received exercise intervention during pregnancy were less likely to develop GDM compared to those who followed the standard prenatal care (OR 0.70, 95%CI 0.52, 0.93; P-value 0.02) [Q 10.08, P-value 0.26; $I^2$ 21% (95%CI 0, 62%]. Studies with low attrition bias also showed a similar result (OR 0.70, 95%CI 0.51, 0.97; P-value 0.03). A protective effect was also supported when analysis was limited to studies including women with low education level (OR 0.55; 95%CI 0.40, 0.74; P-value 0.0001); studies with exercise intervention duration more than 20 weeks (OR 0.54; 95%CI 0.40, 0.74; P-value 0.0007); and studies with a motivation component in the intervention (OR 0.69, 95%CI 0.50, 0.96; P-value 0.03). We could not exclude large variability in study effects because the upper limit of $I^2$ confidence interval was

**Funding:** The authors received no specific funding for this work.

**Competing interests:** The authors have declared that no competing interests exist.

higher than 50% for all analyses. There was no conclusive evidence for small study effects (P-value 0.31).

## Conclusions

Our study might support a protective effect of exercise intervention during pregnancy for high-risk women to prevent GDM. The protective result should be corroborated by large, high quality RCTs.

## Introduction

Gestational diabetes mellitus (GDM) is a multifactorial disorder from the interaction between genetic and environmental risk factors. It is characterized by insulin resistance and decreased pancreatic b-cell function. It is also a risk factor for the future development of type 2 diabetes mellitus [1], and one of the most common diseases during pregnancy [2]. The worldwide prevalence is increasing ranging between 2 and 14% [3]. Women with GDM have an increased risk of obstetric, fetal, neonatal, maternal, and child complications [3–11].

Identified risk factors for GDM include obesity [1, 4–6, 9–13], sedentary lifestyle [10], unbalanced diet [3, 10], socioeconomic factors including low education level [14], ethnicity [2, 4, 10, 15] and family history [4, 10]. Besides interventions for treatment [3], several preventive interventions such as behavioral and lifestyle modifications were evaluated [3, 16].

Previous studies on the effect of exercise intervention were conflicting [4, 5, 13, 17]. Several systematic reviews and meta-analyses [4, 6, 7, 9, 17] showed a significant risk reduction among women in the general population while other studies [8, 10, 18] failed to support risk reduction for GDM. Two recent meta-analyses explored the role of exercise on GDM prevention among high-risk women. One meta-analysis [19], showed no benefit of the interventions, including exercise, compared to placebo; while the other [20] supported a significant GDM risk reduction with exercise during pregnancy among overweight and obese women. However, to our knowledge, there was no systematic approach to evaluate exercise as a single intervention during pregnancy on GDM prevention among high-risk women with any of the risk factors for GDM, and who already received standard prenatal care.

Our study aimed at systematically appraise RCTs that assessed the effectiveness of exercise during pregnancy on the prevention of GDM. We included RCTs on high-risk pregnant women with one or multiple risk factors, which compared exercise to standard prenatal care. We performed meta-analysis with special emphasis on issues of potential biases, and sources of study heterogeneity including both clinical and methodological factors that may account for potential variability in study effects.

## Materials and methods

Our study was registered in the Open Science Framework (OSF) (Registration DOI 10.17605/OSF.IO/23NJS, https://archive.org/details/osf-registrations-23njs-v1). This systematic review was performed according to PRISMA extension for complex interventions guideline [21].

### Search strategy

We searched PubMed, Cochrane Library Central Register of Controlled Trials (CENTRAL), and Scopus (from inception to May 2022). For Pubmed, we used a search strategy including

keywords related to exercise, physical activity, and GDM combined with the Cochrane Collaboration search algorithm for RCTs. We conducted a systematic search on Scopus using the same keywords after excluding articles registered in Pubmed. Finally, we searched CENTRAL including the same keywords related to exercise, physical activity, and GDM. Search algorithms were described in detail in S1 Table. Electronic searches were supplemented by perusal of the references of the retrieved papers as well as the references of review articles. One investigator (GIT) screened all databases. For items considered potentially eligible or unclear, after screening the title and/or abstract, the full text was retrieved. A second investigator (AT) checked on the items that the first investigator (GIT) could not decide. Discrepancies were resolved through consensus. For trials that we could not reach a final decision, or the full text could not be retrieved, we contacted investigators when an e-mail address was available. Two consecutive reminders were also sent to non-responders.

### Eligibility criteria

We selected trials according to PICO (population, intervention, comparator, and outcome) approach. We accepted randomized controlled trials (RCTs) in English that recruited pregnant women at high risk for GDM. Factors that increased pregnant women's risk included at least one of the following: increased BMI [1, 4–6, 9–13], sedentary lifestyle [10], family history [4, 10, 22], previous macrosomia [22], unbalanced diet [3, 10], previous GDM [22], non-white ethnicities [2, 4, 10, 14, 22] and age > 25 years [22]. We considered as eligible trials that assessed interventions of any type of exercise during pregnancy. We accepted trials if women in the comparator group received the standard antenatal care. We considered as eligible trials that reported as outcome the onset of GDM. We accepted all modalities for GDM diagnosis. In case of multiple publications of an RCT with results in different follow-up periods, we accepted the publication including the largest sample. We excluded RCTs that were published at the protocol stage, pilot, or feasibility studies, abstracts from conference proceedings, and RCTs that did not report results on the eligible outcome.

### Data extraction

Two independent researchers (GIT and KP) extracted the data. Discrepancies were resolved with consensus, and the participation of a third arbitrator (AT) where necessary. The Cohen kappa coefficient with 95% confidence interval (95% CI) was used to evaluate the agreement between the two investigators who independently extracted the data.

Extracted items included the name of first author, year of publication, country, whether the study was a cluster RCT, number of participating centers, study duration, drop-out rate, sample size, factors related to high risk for GDM in the participants, women's mean age, and number of participating women with low level of education if reported. We also recorded the type of intervention and the care that women in the comparator group received. For assessing the completeness of exercise intervention reporting, we used the CERT (Consensus on Exercise Reporting Template) tool for complex interventions [23]. CERT was proposed to improve reporting of exercise intervention programs in clinical trials. It included 16 items allocated in 7 categories, i.e., materials, provider, delivery, location, dosage, tailoring, and to what extent the exercise intervention was delivered and performed as planned [23]. In addition, we extracted potential side-effects /adverse events that were reported for intervention, and comparator arm. Finally, we recorded the number of GDM events as the outcome, separately in the experimental and the control arm. We also captured information on the method used in each study for the diagnosis of GDM.

### Quality assessment of the studies and rating of overall evidence

We used the risk of bias tool proposed by the Cochrane Collaboration [24] for quality assessment of eligible RCTs. Two independent researchers (GIT and KP) extracted the data on quality assessment. Discrepancies were resolved with consensus, and the participation of a third arbitrator (AT) where necessary. In addition, we used the Grading of Recommendations, Assessment, Development and Evaluation tool (GRADE) for rating the overall evidence [25] (GRADEpro, Version 3.6.1. McMaster University, 2011)".

### Statistical analysis

To combine the events of GDM, we performed both fixed effects and random effects model (REM) meta-analyses. In case that large heterogeneity could not be excluded, we reported the REM results (odds ratio with 95% CI) [26]. Heterogeneity was evaluated with Cochran's Q statistic (statistically significant for $P < 0.10$); and it was quantified with the $I^2$ metric (low, moderate, large, very large for values of $<25$, $25$–$49$, $50$–$74$, $>75\%$, respectively) [27]. The main analyses included all available data. We performed separate analyses limited to studies where increased BMI was included in as a risk factor for GDM, and studies that did not consider BMI; studies where the percentage of participating women with low level education was more than 5%; studies that evaluated an intervention delivered individually, and studies that evaluated an intervention delivered in a group; trials that included a motivation component in the intervention, and trials that did not include motivation; studies with an intervention duration more than 20 weeks, and studies with an intervention duration up to 20 weeks. We also performed meta-regression analyses on GDM OR. The effect of baseline risk, and study duration were included individually as covariates in the meta-regressions. For each meta-regression, the slope coefficient with the standard error (SE), the permutation-based P-value (as suggested by Higgins and Thompson [28] and the $tau^2$ were reported. Publication bias was evaluated via the visual analysis of funnel plot, showing a symmetrical inverted funnel in the absence of bias [29]. To further investigate potential asymmetry due to publication bias, we performed the statistical Egger's test [30]. We also performed separate analyses for studies with low detection bias (studies reporting blinding of outcome assessors); and for studies with low attrition bias (studies with less than 20% of participants lost in follow-up). The level of significance for all analyses, except for Cochran's Q statistic, was set at P-value $< 0.05$. For our analyses, we used SPSS 22.0 (SPSS, Inc., Chicago, Illinois, USA), Stata Statistical Software 10.1 (Stata, College Station, TX, USA), and Review Manager 5.4 (Cochrane Collaboration, UK).

## Results

### Eligible studies

Our search yielded 1566 items (582 in PubMed, 290 in Scopus, and 694 in CENTRAL). We excluded 268 as duplicated. Out of the 1298 remaining items, we excluded 1260 as non-relevant based on the title, or abstract. Thus, we retrieved 37 papers in full text. Out of the 38 articles, we excluded 29; one paper reported a pilot study; 8 studies did not include an eligible population; 7 studies included a non-eligible intervention; and 13 trials did not report the onset of GDM as an outcome. Finally, we included 9 published RCTs as eligible for our study (Fig 1).

### Characteristics of eligible studies

Eligible studies were published from 2012 to 2017. Four RCTs [13, 31–33] were conducted in Europe (one study in Netherlands, one in Spain, one in Norway, and another one in Ireland);

PRISMA 2020 flow diagram for new systematic reviews which included searches of databases

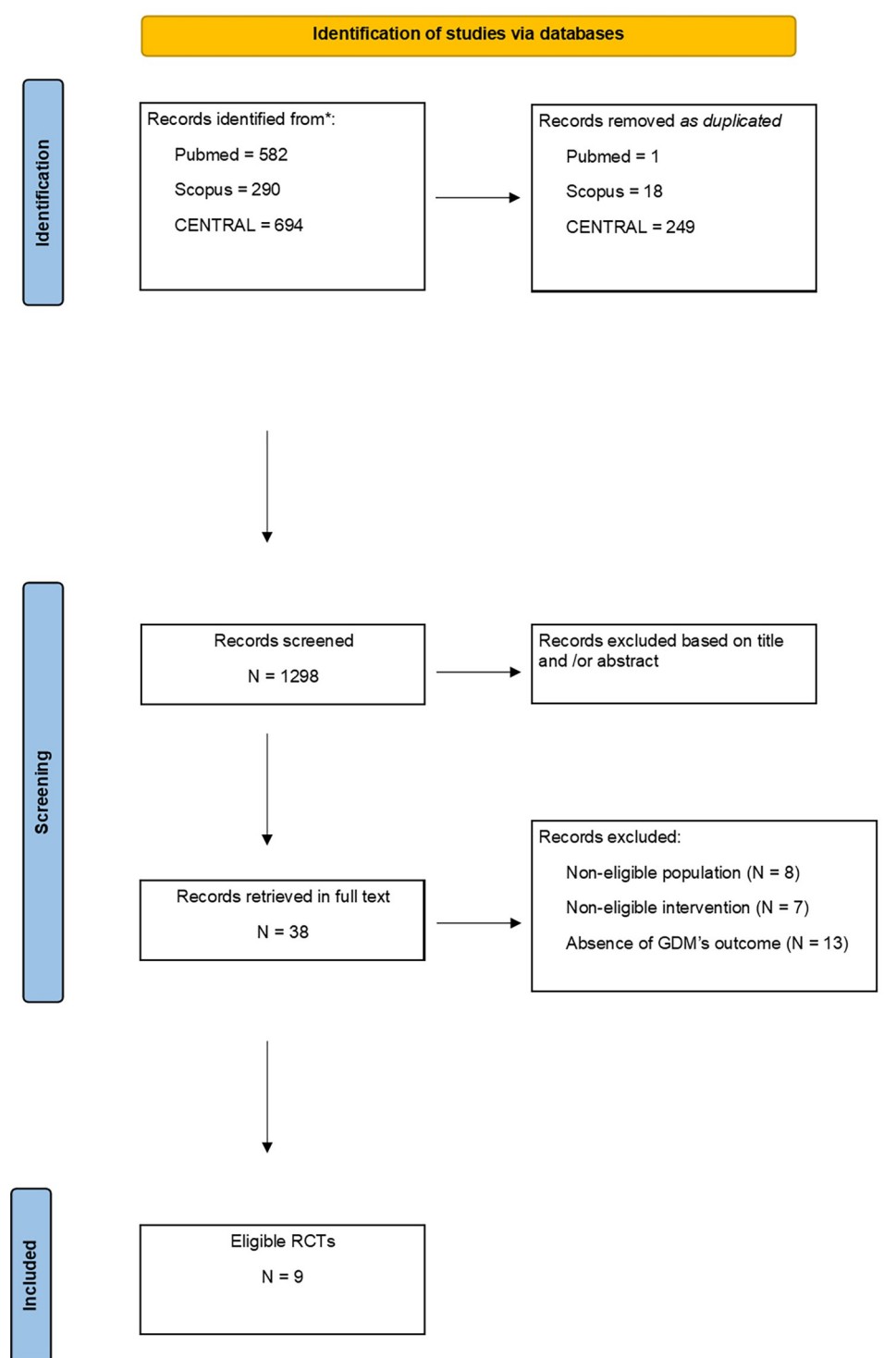

**Fig 1. Flow chart of study selection process.**

**Table 1. Characteristics of eligible studies.**

| First author, publication year | Country | Number of participated centers | Study duration, mo | Drop-out rate n (%) |
|---|---|---|---|---|
| Oostdam, 2012 | Netherlands | 5 | 48 | 22 (18.2) |
| Price, 2012 | USA | 1 | 45 | 29 (31.9) |
| Barakat, 2013 | Spain | 1 | 40 | 82 (16) |
| Nobles, 2015 | USA | 1 | 60 | 39 (13.4) |
| Seneviratne, 2015 | New Zealand | 1 | 19 | 1 (1.3) |
| Guelfi, 2016 | Australia | 1 | 37 | 3 (1.7) |
| Krohn Garnæs, 2016 | Norway | 1 | 22 | 17 (18.7) |
| Wang, 2017 | China | 1 | 20 | 35 (11.7) |
| Daly, 2017 | Ireland | 1 | 41 | 2 (2.3) |

mo, months

two studies was conducted in Oceania [34, 35], (one study in New Zealand, and one in Australia), two in USA [2, 36], and another one [5] in China (Table 1). All trials used the participant as the randomization unit and had a parallel design. One study [31] was multi-centered (five participating centers). The duration of the trials ranged from 19 to 60 months. The drop-out rate was < 20% for all studies, except for one study [36] that was 31.9% (Table 1).

A total of 1,738 (866 in intervention, and 872 in control group) high-risk women for GDM participated in the eligible trials. Six studies included overweight, and obesity as risk factors [2, 5, 13, 31, 32, 34]. Additional risk factors included history of GDM in three studies [2, 31, 35], history of type 1 and 2 diabetes mellitus in first- and second-degree relatives in two studies [2, 31], history of macrosomia in one study [31], and previously sedentary lifestyle in two studies [33, 36] (Table 2). Mean age ranged from 24.9 to 37.7 years for women in the intervention

**Table 2. Characteristics of participating women in the eligible studies.**

| First author, publication year | Sample size (intervention / control) | Risk factors for GDM | Mean age (SD), yr intervention / control | Low education level, n (%) intervention / control |
|---|---|---|---|---|
| Oostdam, 2012 | 121 (62 / 59) | Obese (body mass index, BMI ≥ 30) or overweight (BMI ≥ 25) AND at least one of the three following characteristics: (1) history of macrosomia (offspring with a birthweight above the 97th percentile of gestational age); (2) history of GDM; or (3) first-grade relative with T2D | 30.8 (5.2) / 30.1 (4.5) | 16 (34) / 17 (34.7) |
| Price, 2012 | 91 (43 / 48) | Previously sedentary women; no aerobic exercise more than once per week for at least the past 6 months | 30.5 (5) / 27.6 (7.3) | ND |
| Barakat, 2013 | 510 (255 / 255) | Previously sedentary women; not exercising more than 20 min on more than 3 days/week | 31 (3) / 31 (4) | 54 (25.7) / 75 (34.4) |
| Nobles, 2015 | 290 (143 / 147) | Overweight or obese (pre-pregnancy BMI ≥ 25 kg/m²) with a family history of DM or a diagnosis of GD in prior pregnancy, defined according to the ADA criteria | Range 18–40 | 26 (22) / 31 (27) |
| Seneviratne, 2015 | 75 (38 / 37) | Pre-pregnancy BMI ≥ 25 kg/m² | ND | ND |
| Guelfi, 2016 | 172 (85 / 87) | Pregnant women with a history of GDM in a previous pregnancy | 33.6 (4.1) / 33.8 (3.9) | ND |
| Krohn Garnæs, 2016 | 91 (46 / 45) | Pre-pregnancy BMI ≥ 28 kg/m² | 31.3 (3.8) / 31.4 (4.7) | 1 (2) / 3 (7) |
| Wang, 2017 | 300 (150 / 150) | Pre-pregnancy BMI ≥ 24 kg/m² | 32.1 (4.6) / 32.5 (4.9) | 31 (21) / 40 (27) |
| Daly, 2017 | 88 (44 / 44) | BMIs at their first prenatal visit of 30 or greater | 30.0 (5.1) / 29.4 (4.8) | ND |

GDM, gestational diabetes mellitus; SD, standard deviation; yr, years; BMI, body mass index; T2D, type 2 diabetes; DM, diabetes mellitus; ADA, American Diabetes Association; ND, no data

group, and from 20.3 to 37.7 years for women in the control group (Table 2). One study [2] reported only the age range (18 to 40 years) (Table 2). Percentage of women with low education level ranged from 2% to 34% in the intervention group, and from 7% to 34.7% in the control group. Four studies [32, 34–36] did not report data on participants' education level (Table 2).

Interventions evaluated several exercise programs with the use of various equipment (Table 3). Five [2, 13, 31–33] out of the nine studies also included a motivation component in the intervention. Providers included physiotherapists in two studies [13, 31], health educators in one trial [2], exercise physiologist in two RCTs [34, 35], researchers in three trials [5, 32, 36], and fitness specialist with the assistance of an obstetrician in one study [33]. Three trials [31, 34, 35] delivered the intervention individually; one [36] both in group and individually; three studies [13, 32, 33] delivered the intervention in group; and two studies did not report relevant data [2, 5] (Table 3). Seven trials [5, 13, 31–33, 35, 36] evaluated a supervised intervention (Table 3). The duration of the intervention was more than 20 weeks in five trials [5, 13, 31, 33, 36] (Table 3).

**Table 3. Interventions in eligible studies.**

| First author, publication year | Intervention brief description | Provider | Type of intervention | Duration of intervention |
|---|---|---|---|---|
| Oostdam, 2012 | Warming-up such as slow cycling, individualized program of aerobic and strength exercises, cool down. | Physiotherapist | Individualised; supervised | From 15 wks of gestation to delivery |
| | Equipment: cycle ergometers, treadmills, cross-trainers, stationary rowing machines, free weights, accelerometer. | | | |
| | Motivation component: Information on the benefits for mother and child at the start and during the intervention. | | | |
| Price, 2012 | Aerobic training 4 times per week, 3 times at moderate intensity as a group, consistent with exercise guidelines of the ACOG. Also, walk individually once weekly. | Researchers | Both as a group and individually; supervised | From 12–14 week of gestation to 36 week of gestation or to delivery if participants wished |
| | Equipment: Treadmills, elliptical trainers, stationary bicycles, weight machines, exercise balls. | | | |
| | Motivation: not included | | | |
| Barakat, 2013 | Aerobic dance activities of 3–4 min with 1 min breaks, moderate-intensity resistance exercises lasted 25–30 min, warm-up and cool-down period both of 10–12 min duration with standards of the American College of Obstetricians and Gynaecologists. | Fitness specialist with the assistance of an obstetrician | In a group; supervised | From weeks 10 to 12 of pregnancy to the end of the third trimester (weeks 38–39) |
| | Equipment: Bar-bells, therabands, heart rate monitor | | | |
| | Motivation: All sessions were accompanied with music, and were performed in an airy, well-lighted exercise room at the Hospital. | | | |
| Nobles, 2015 | 30 minutes or more of moderate-intensity physical activity on most days of the week. Specific activities self-selected and including dancing, walking, and yard work. | Health educators | ND | 10 wks on avarage |
| | Equipment: Digital pedometer. | | | |
| | Motivation: Booster telephone calls and tip sheets mailed. | | | |
| Seneviratne, 2015 | Cycling home based moderate-intensity exercise sessions. Each exercise session included a 5-minute warm-up and cool-down period at low intensity. Frequency varyed between three and five sessions per week, and duration between 15 and 30 minutes per session, according to stage of pregnancy. | Exercise physiologist | Individualised; unsupervised | From 20 to 35 weeks of gestation |
| | Equipment: Magnetic stationary bicycles, heart rate monitors | | | |
| | Motivation: not included | | | |

*(Continued)*

**Table 3.** (Continued)

| First author, publication year | Intervention brief description | Provider | Type of intervention | Duration of intervention |
|---|---|---|---|---|
| Guelfi, 2016 | 5-minute warm up of pedaling, 5-min periods of continuous moderate-intensity cycling alternating with 5-min periods of interval cycling, 5-minute cool down followed by light stretching. | Exercise physiologist | Individualised; supervised | 14 wks |
| | Equipment: Upright cycle ergometer, accelerometer. | | | |
| | Motivation: not included | | | |
| Krohn Garnæs, 2016 | Three times a week 35 minutes of moderate-intensity endurance exercise and 25 minutes of strength training. Determination of the endurance exercise at 80% of the maximum capacity, according to the Borg scale 12–15. | Physiotherapist | In a group; supervised | From 12th-18th gestational week to delivery |
| | Equipment: Treadmill. | | | |
| | Motivation: Motivational interview session, either individually or in a group and encouragement to compare their own weight gain with the recommended. | | | |
| Wang, 2017 | Exercise at the beginning of the intervention at the lower calculated limit, based on the maximum predicted heart rate for age, progressively increased with the progress of the program, at least 3 days a week. | Researchers | Supervised | $27 \pm 2$ wks |
| | Equipment: Stationary bike. | | | |
| | Motivation: not included | | | |
| Daly, 2017 | 10-minute warm-up, 15–20 minutes of resistance or weights, 15–20 minutes of aerobic exercises, and a 10-minute cool-down. | Researchers | In a group; supervised | 13 4/7 $\pm$ 1 2/7 wks of gestation |
| | Equipment: Weights. | | | |
| | Motivation: Goal-setting and journaling of varied classes each day to maintain interest. Women in the intervention arm also received an invitation to a secret Facebook group to create a sense of community among participants, to share healthy lifestyle advice, and to improve compliance with the exercise intervention. | | | |

wks, weeks; ND, no data

### Reporting of exercise intervention in eligible studies

Based on CERT [23], we captured the number of studies with inadequate reporting of the description of the exercise intervention (S2 Table). There was no trial that provided with adequate information for exercise reproduction. Six [2, 5, 31–33, 36] out of nine papers did not provide any information on the content of home program component. Six trials [5, 13, 31, 33–35] did not adequately report on non-exercise components. Generally, all trials provided information on the exercise components, and the necessary equipment; on the provider, and the supervision of the intervention; as well as on adherence, on potential side-effects /adverse events, and on dosage.

### Effectiveness and safety of exercise during pregnancy

GDM was diagnosed by measuring fasting blood glucose, hemoglobin A1c, or by an oral glucose tolerance test (Table 4). A total of 374 (24.8%) [160 (21.4%) in intervention, and 214 (28.1%) in control group] developed GDM among the 1,508 high-risk women analysed for GDM outcome (Fig 2). When the nine trials were combined, there was no between study heterogeneity (Q 10.08, P-value 0.26). However, we could not exclude large variability (upper

**Table 4. Tests used for the diagnosis of GDM, and reported side effects /adverse events in the eligible RCTs.**

| First author, publication year | Diagnostic test for GDM | Side effects /adverse events (intervention /control) |
|---|---|---|
| Oostdam, 2012 | FBG, HbA1c | none reported |
| Price, 2012 | 50-g 1hr OGTT | anxiety with exercise (1 / 0); history of preterm pregnancy (1 / 0); pain from leiomyomas (1 / 0) |
| Barakat, 2013 | 75-gr OGTT 2hr | premature labour (5 / 3); pregnancy-induced hypertension (5 / 4); persistent bleeding (3 / 0); molar pregnancy (0 / 3) |
| Nobles, 2015 | 50-gr 1hr OGTT | developed medical contraindication (3 / 1); miscarriage or termination (1 / 2) |
| Seneviratne, 2015 | 75-gr OGTT FBG and / or 2hr | none reported |
| Guelfi, 2016 | FBG, 75-gr OGTT 2hr, or both | pregnancy loss (1 / 2) |
| Krohn Garnæs, 2016 | FBG or 120-min BG | none reported |
| Wang, 2017 | 75-gr 2hr OGTT | cervical length < 25 mm (1 / 5); other* |
| Daly, 2017 | 75-gr 2hr OGTT | none reported |

GDM, gestational diabetes mellitus; OR, odds ratio; CI, confidence interval; FBG, fasting blood glucose; HbA1c, hemoglobin A1c; min, minutes; BG, blood glucose; OGTT, oral glucose tolerance test; ND, no data

* other included 4 side effects / adverse events (intervention / control): ankle sprain (1 / 0); low-lying placenta (1 / 0); fetal death in utero (0 / 1); malformation (0 / 1)

limit for $^2$ > 50%) in study effects due to real study differences [$I^2$ 21% (95%CI 0, 62%)]. Thus, random effects estimates would be more appropriate for data synthesis and fixed effects estimates were not presented. Women who received exercise during pregnancy were on average less likely to develop GDM compared to women who followed only the standard prenatal care (OR 0.70, 95%CI 0.52, 0.93; P-value 0.02) (Fig 2).

The summary odds ratio showed also a significant effect when analyses were limited to studies with more than 5% of the participating women reporting a low education level (OR 0.55, 95%CI 0.40, 0.74; P-value 0.0001); studies reporting the use of a motivation component in the intervention (OR 0.69, 95%CI 0.50, 0.96; P-value 0.03); and studies that evaluated an intervention with duration more than 20 weeks (OR 0.54, 95%CI 0.40, 0.74; P-value 0.0001). However, the test of difference was significant only for the subgroup analysis based on exercise duration (studies with exercise duration more than 20 weeks vs. studies with duration up to 20 weeks; P-value = 0.02) (S3 Table). In sensitivity analyses, the summary odds ratio remained

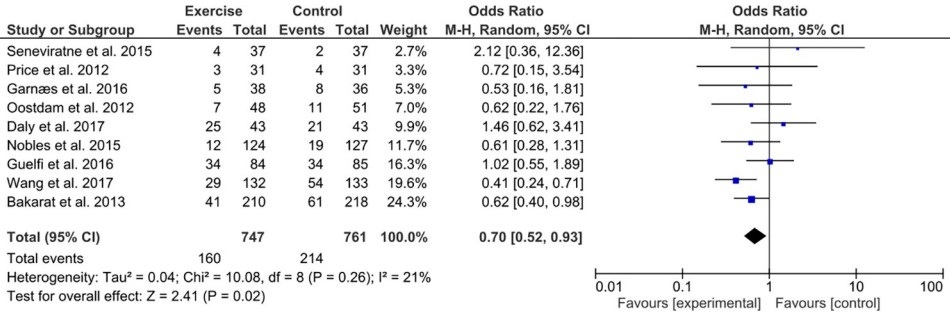

**Fig 2. Exercise intervention and the risk of gestational diabetes mellitus in high-risk pregnant women.** Each study is shown by an odds ratio (OR) estimate, along with 'whiskers' corresponding to its 95% confidence interval (95% CI). Studies are ordered according to the weight they contributed to the meta-analysis. The summary OR by random effects calculations is also shown.

statistically significant when studies were limited to those with a low attrition bias (OR 0.70, 95%CI 0.51, 0.97; P-value 0.03). We could not exclude large variability in study effects due to real study differences for all subgroup and sensitivity analyses (S3 Table). Thus, even statistically significant effects should be interpreted with caution because the true differences in effects across studies might be due to unidentified or unexplained underlying factors. Meta-regression analyses with baseline risk, and study duration as covariates did not show a statistically significant effect on the summary OR (S4 Table).

Pregnancy-induced hypertension was the most frequently reported adverse event. Four trials [13, 31, 32, 34] reported that there was no adverse event (Table 4).

## Quality of reporting, potential bias, and quality of evidence

There was good agreement between the two independent researchers [Cohen k 91.4% (95% CI 82.8%, 100%; P-value <0.001)]. Based on the overall risk, four trials were judged to raise some concerns because they failed to report specific quality domains. Specifically, two trials [2, 35] did not provide information on participants and personnel blinding, and on blinding of outcome assessors; and two studies [13, 31] did not provide information on participants and personnel blinding only. Three RCTs were judged to be at high risk of bias. One of them [36] did not provide information on participants and personnel blinding, and on blinding of outcome assessors; in addition, it reported a drop-out rate at 31.9%. The other two trials [5, 32] were unblinded for participants and personnel; one of them [5] was also unblinded for outcome assessors (Table 5).

Based on the funnel plot assessment, there was variation in the standard error of the studies. However, small studies were reasonably closely distributed around the summary effect estimate [29] (Fig 3). Egger's test of small study effects had a P-value of 0.31, and thus, it was not fully conclusive.

Five out of the 9 studies had potential performance, detection, or attrition bias. The other 4 studies were unclear about blinding. Overall, moderate quality of evidence showed that exercise during pregnancy for the population of women with high risk for GDM may have benefit when compared to standard prenatal care in reducing the risk of GDM (S5 Table).

**Table 5. Quality of reporting for eligible studies.**

| First author, publication year | Random sequence generation (selection bias) | Allocation concealment (selection bias) | Blinding of participants and personnel (performance bias) | Blinding of outcome assessment (detection bias) | Incomplete outcome data (attrition bias) | Selective reporting (reporting bias) | Other bias |
|---|---|---|---|---|---|---|---|
| Oostdam, 2012 | L | L | ? | L | L | L | L |
| Price, 2012 | L | L | ? | ? | H | L | L |
| Barakat, 2013 | L | L | H | L | L | L | L |
| Nobles, 2015 | L | L | ? | ? | L | L | L |
| Seneviratne, 2015 | L | L | H | L | L | L | L |
| Guelfi, 2016 | L | L | ? | ? | L | L | L |
| Krohn Garnæs, 2016 | L | L | ? | L | L | L | L |
| Wang, 2017 | L | L | H | H | L | L | L |
| Daly, 2017 | L | L | H | L | L | L | L |

H, high risk; L, low risk;?, unclear

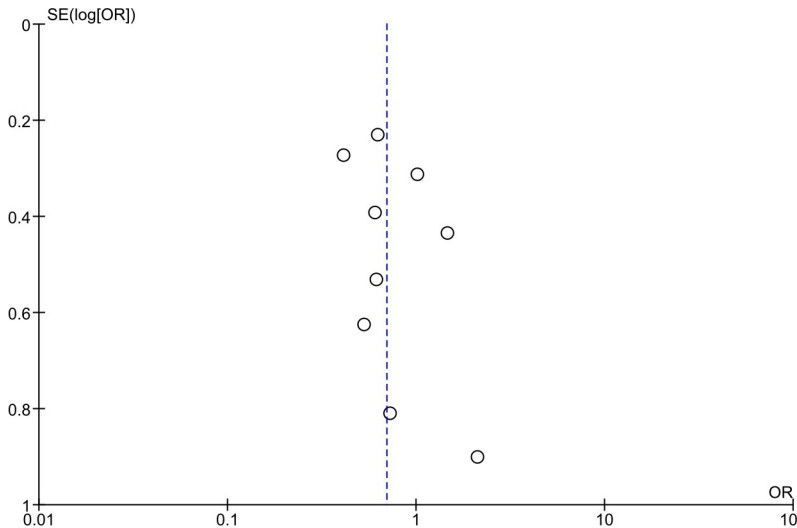

**Fig 3. Funnel plot including all studies comparing an exercise intervention vs. standard prenatal care for gestational diabetes prevention among pregnant high-risk women [P-value 0.68 in the weighted regression of ln (OR) against the standard error].**

## Discussion

Our study showed that on average an exercise intervention during pregnancy may have a beneficial effect in preventing high-risk pregnant women from developing GDM. There was no significant between study heterogeneity. However, we noticed that a large variability in study effects could not be excluded. A potential beneficial effect was also supported when analyses were limited to studies with more than 5% of the participating women reporting a low education level; studies reporting the use of a motivation component in the intervention; and studies that evaluated an intervention with duration more than 20 weeks. Subgroup and sensitivity analyses did not identify a clinical or methodological factor that may explain for the potential large variability.

Our meta-analysis supported the possibility that specific exercise programs during pregnancy may decrease the GDM incidence. Exercise programs should follow guidelines for designing complex interventions [21]. Based on CERT [23], reporting of several intervention characteristics was missing. The study [5] with a significant decrease in GDM incidence did not provide data on whether the intervention was in group or applied individually; on any motivation strategies, on the content of home exercise, and on other non-exercise components; and on whether the exercise intervention was individually tailored or not. Thus, it may not be feasible for this intervention to be reproduced in future trials. Previous studies on complex interventions also showed inadequate reporting [4, 6–10, 17, 18].

Other interventions such as diet, supplements, and medications were evaluated for GDM prevention. Results regarding these outcomes also need to be scrutinized. Some of these interventions may be important but spurious effects due to various biases may be affecting these trials as well. For general population, some meta-analyses assessing exercise interventions with or without a diet component showed also statistically significant GDM risk reduction [4, 6, 7, 9, 17]. However, other studies [8, 10, 18, 19] did not support similar results. In line with our study, a meta-analysis that evaluated exercise among overweight or obese women showed a reduction in GDM incidence [20]. Another meta-analysis [37] that evaluated the effect of different types of exercise and metformin for pregnancy outcomes in overweight and obese

pregnant women, showed a reduced risk for GDM with aerobic exercise. However, our subgroup analysis limited to studies that included high-risk women based on the BMI criterion, did not show a significant effect of exercise intervention on GDM incidence. Compared to previous studies, our meta-analysis followed a more pragmatic approach for population eligibility including not only studies with pregnant women with increased BMI but also studies with pregnant women with other risk factors for GDM. Based on our subgroup analysis, future research on exercise interventions with adequate duration among pregnant women might be promising. However, this result should be interpreted cautiously since a large variability in study effects could not be excluded.

Several modifiable factors as well as non-modifiable factors may contribute to GDM. Obese women had twice the risk for GDM as compared to women with normal body weight [5]. Elevated pre-pregnancy BMI is associated with complications during pregnancy, regardless of GDM onset [1, 14, 16]. A cost-effectiveness study [38] showed that promoting healthy eating and physical activity was the preferred strategy for limiting weight gain during pregnancy. However, the exact intervention components that may lead to clinically significant risk reduction are yet to be determined. Previous studies supported that women during pregnancy showed low motivation to change their lifestyle [11]. Our subgroup analysis limited to studies that included a motivation component in the intervention also supported a significant effect of exercise intervention on GDM incidence. Therefore, exercise interventions may include a behavior change component. They may also address social determinants of health including education level to improve literacy, and access to health care services in addition to biological factors, and the right timing for women to start the intervention to prevent GDM.

Our findings may support on average a protective effect of exercise intervention during pregnancy for GDM prevention among women with GDM risk factors. However, both for the main analysis and for the subgroup and sensitivity analyses, potential large true differences in effects among studies could not be excluded. There may be additional unidentified or unexplained underlying factors that may account of the differences in effects. Future large, good quality trials recruiting pregnant women of low education level and evaluating an exercise intervention with satisfactory duration need to adequately report on the intervention characteristics to allow for evaluating potential frequency-response relationship between exercise and GDM risk reduction [12]. Additionally, they need to provide with adequate description of the exercise intervention programs for their accurate reproduction [23]. Motivation techniques for participants to complete the intervention, intensive monitoring to minimize losses to follow up that are not due to miscarriage, premature delivery, or fetal death in utero, and procedures that enhance fidelity are prerequisites for adequately implementing exercise interventions. Additional efforts to ensure blindness both of participants and researchers are imperative to support robustness of the results. In previous trials, investigators found it difficult to double-blind RCTs due to the nature of the intervention [3, 11]. Previous results on diet interventions to prevent GDM were also heterogeneous [12, 13]. Future trials assessing interventions including multiple components, i.e., diet, exercise, behavioral counseling, and social support, are needed to provide with definitive answers on their benefit and sustainability [10, 18].

Our study had several limitations. We included only studies that evaluated interventions initiated during pregnancy; therefore, our findings cannot be generalized to exercise interventions that may begin before pregnancy. However, this limited the heterogeneity of the duration of intervention among studies. We included RCTs that recruited only high-risk women; and therefore, our results cannot be generalized to general population. However, we considered as high-risk not only women who were overweight or obese but also women with other risk factors including ethnicity, medical, and family history, and sedentary lifestyle. By broadening

the criteria, we tried to achieve a pragmatic approach of the population included in our work. Searching for grey literature might have identified additional studies; however, unpublished results would still have remained unknown.

## Conclusion

As a conclusion, our study may support a beneficial effect of exercise interventions during pregnancy in addition to standard antenatal care for preventing GDM among high-risk women. Furthermore, a protective effect for specific population subgroups, i.e., women with low education level, and for interventions with specific characteristics, i.e., with more than 20 weeks duration, and with motivational strategies cannot be excluded. Future large, good quality studies focusing on specific women populations, and evaluating interventions with adequate duration are necessary.

## Supporting information

**S1 Table. Search strategy.**
(DOCX)

**S2 Table. Evaluation of the exercise intervention based on Consensus on Exercise Reporting Template (CERT) tool.**
(DOCX)

**S3 Table. Subgroup and sensitivity analyses.**
(DOCX)

**S4 Table. Meta-regression results for GDM OR.**
(DOCX)

**S5 Table. GRADE evaluation of overall evidence.**
(DOCX)

**S1 File. PRISMA checklist.**
(DOCX)

## Author Contributions

**Conceptualization:** Georgios I. Tsironikos, Athina Tatsioni.

**Data curation:** Georgios I. Tsironikos, Konstantinos Perivoliotis, Athina Tatsioni.

**Formal analysis:** Georgios I. Tsironikos, Athina Tatsioni.

**Funding acquisition:** Elias Zintzaras.

**Investigation:** Georgios I. Tsironikos, Konstantinos Perivoliotis, Athina Tatsioni.

**Methodology:** Georgios I. Tsironikos, Konstantinos Perivoliotis, Alexandra Bargiota, Elias Zintzaras, Chrysoula Doxani, Athina Tatsioni.

**Supervision:** Georgios I. Tsironikos, Athina Tatsioni.

**Writing – original draft:** Georgios I. Tsironikos, Athina Tatsioni.

**Writing – review & editing:** Georgios I. Tsironikos, Konstantinos Perivoliotis, Alexandra Bargiota, Elias Zintzaras, Chrysoula Doxani, Athina Tatsioni.

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
