## [Decision Letter · Decision Letter 0]

17 Jan 2022

PONE-D-21-21724Effectiveness of exercise intervention during pregnancy on high-risk women for gestational diabetes mellitus prevention: a meta-analysis of published RCTsPLOS ONE

Dear Dr. Tatsioni,

Thank you for submitting your manuscript to PLOS ONE. After careful consideration, we feel that it has merit but does not fully meet PLOS ONE’s publication criteria as it currently stands. Therefore, we invite you to submit a revised version of the manuscript that addresses the points raised during the review process.

We look forward to receiving your revised manuscript.

Kind regards,

Carsten Bogh Juhl, PhD

Academic Editor

PLOS ONE

https://journals.plos.org/plosone/s/file?id=ba62/PLOSOne_formatting_sample_title_authors_affiliations.pdf”.

Additional Editor Comments:

Dear author – thank you for your systematic review and meta-analysis on an important health problem. However some issues remain as pointed out by the reviewers. Further I have some additional issues that may need to be addressed before publication.

In the literature search was the MeSH terms used or was the search just performed as text words. The use of the filter – randomized controlled trial – may be less sensitive than the Cochrane highly sensitive filter for identifying RCT (these issues need to be addressed as limitation)

The study was not protocolized in either open science framework (www.osf) nor in PROSPERO. This is a severe limitation and need to be addressed too.

The study was reported according to the PRISMA guidelines – using the PRISMA flowchart may improve the readability and the quality of the flowchart. Further the reporting in the flowchart is inconsistent – 780 hits was retrieved and 72 was duplicates – but only 474 was screened – and 639 were excluded based on title and abstract. Further some difference between text and flowchart are seen (73 or 72 duplicates).

The author used Odds Ratio (OR) for pooling the results – even though OR showed more extreme results compared to Relative Risk (RR) and all included studies are RCT making an meta-analysis on RR possible. Presenting the results in RR may easy the interpretability of the results for clinician. Further the plots are quite blurry (both the forest plot and the flowchart).

The author stated that they has especial focus on explaining the heterogeneity - however why is the only analysis for investigating heterogeneity Eggers test for small study bias – what about the amount of exercise – the duration of the exercise intervention – the risk of bias – et cetera. These analysis can be performed comparing subgroups or investigated in a meta-analysis.

Finally the GRADE approach should be used for evaluating the overall evidence (based on study limitation, inconsistency, indirectness, impression and publication bias)

Best Associate Professor Carsten Juhl

Reviewers' comments:

Reviewer's Responses to Questions

**Comments to the Author**

1. Is the manuscript technically sound, and do the data support the conclusions?

Reviewer #1: Yes

Reviewer #2: Partly

2. Has the statistical analysis been performed appropriately and rigorously? 

Reviewer #1: Yes

Reviewer #2: Yes

3. Have the authors made all data underlying the findings in their manuscript fully available?

Reviewer #1: Yes

Reviewer #2: Yes

4. Is the manuscript presented in an intelligible fashion and written in standard English?

Reviewer #1: Yes

Reviewer #2: No

5. Review Comments to the Author

Reviewer #1: Εffectiveness of exercise intervention during pregnancy on high-risk women for gestational diabetes mellitus prevention: a meta-analysis of published RCTs

In this meta-analysis of published RCTs on exercise in pregnant women at risk of GDM, the authors find a significant benefit of exercise in preventing GDM but prudently infer cation to their results due to huge heterogeneity of the ten included RCT studies. The meta-analysis contributes to the existing body of literature on the topic.

The strengths of the study include a strong adherence to reporting guidelines such as PRISMA and standardized methods such as CERT and tools proposed by the Cochrane Collaboration. The method section is rigorous and well written. The authors have a pragmatic approach to the risk factor for GDM in the included population which is well elucidated. The weaknesses of the included studies are clarified and well put in the context of the meta-analysis.

The shortcoming of the manuscript is the difficulty of addressing the impact of the findings and viewing the results in the context of similar studies. What is the impact of this meta-analysis compared to the other meta-analysis on the topic?

There are also a few more questions that require addressing:

Abstract page 3: The last four lines of the Results section are repeated in the conclusion. Consider omitting the four lines in the results section to include more results from the meta-analysis.

The first line of the conclusion should address the aim of the study.

Introduction page 4: The authors mention several risk factors for GDM but in the research question it is not clear how they define “high-risk pregnant women”. Do the women have 1, 2, 3 4 or all of the mentioned risk factors to be at high risk. The definition of “high-risk pregnant women” should be elucidated already in the introduction?

Methods section (Eligibility Criteria) page 5: Ethnicity as a risk factor could warrant elaboration since it is not as intuitive as the other risk factors.

Discussion section page 15-16: To strengthen the discussion section, it would be beneficial with a more direct comparison of your study to [ref 18] and [ref 19].

[ref 18]: This meta-analysis found no effect of physical exercise on GDM. Why is their conclusion different to yours? Differences in eligibility criteria for included studies or differences in the characteristics of participating women in the eligible studies?

[ref 19]: This meta-analysis also finds a beneficial effect of exercise but with fewer reservations about their result. Is it only because of prenatal exercise or small sample sizes in included studies? Be kind to elaborate – what does the author's scepticism consist of?

Discussion section page 17: The authors have to further elaborate on how this study adds to what is already known on exercise to prevent GDM apart from identifying the presence of publication bias. However, the authors succeed in suggesting improvements in future RCT such as an adequate description of the exercise intervention and motivational techniques for participants to complete the intervention.

Reviewer #2: This article addresses the preventing effect of exercise among women with high risk of gestational diabetes mellitus during pregnancy. Which is an important topic.

There are some areas in this article, that needs to be addressed/clarified. In general, the writing quality needs to be improved to a higher and more professional level.

The paper is with sufficient details.

Abstract:

Overall the abstract is relevant and capture the articles.

The phrase “between-study heterogeneity was estimated” is used multiple times during the article.

Ex. page 3: “There was significant between study heterogeneity [Q 24.45, P-value 0.004; I2 = 63% (95%CI 27%, 81%)]”.

It seems like you have mixed two expressions, heterogeneity is between study variance above the expected (chance), so you have to decide to write between study variance or heterogeneity.

Besides that, one should separate the values, so one cite the Q and I value for themself. because I is the proportion of the total variation due to inconsistency, and Q is saying if the variation is random (If Q is significant (p < 0.05) => heterogeneity).

Introduction:

In the introduction it would be relevant to mention education levels impact on GDM, since it is mentioned in table 2 later.

On page 4 it is stated that “We performed meta-analysis with special emphasis on issues of potential biases and sources of heterogeneity between studies.” Here it would be good to analyze what the heterogeneity is, and not only if there is publication bias.

Methods:

Some parts of the Methods section are not adequately explained

In the introduction one writes “Identified risk factors for GDM include obesity [1,4-6,9-13], sedentary lifestyle [10], unbalanced diet [3,10], ethnicity [2,4,10,14] and family history [4,10] “and then in the Eligibility Criteria you elaborate, but it is highly recommended to elaborate on ethnicity as well.

Besides that, sedentary lifestyle and unbalanced diet is mentioned as a risk factor, but not used as an eligibility Criteria. What are your thoughts about that specific? A small comment about this would be good to include.

In the data extraction on page 6 you mention that you extract level of education, but its not used it for anything, please think about why you extract it.

Under the quality assessment of the studies its written that: “We used the risk of bias tool proposed by the Cochrane Collaboration for quality assessment of eligible RCTs “. But it’s not used for anything. Its only too state the findings, but not what it means for the interpretation of the study as a whole.

It’s not mentioned how many authors asses the quality of the studies, this is first mentioned during the presentation of the results, consider to mention this in method section instead of in the results. Besides that, you should mention what outcome you asses.

Under statistical Analysis on page 6 you write that “Heterogeneity was evaluated with Cochran’s Q statistic (statistically significant for P < 0.10)” and later you write “The level of significance was set at P < 0.05”. Hence, it would be good to clarify what significant level that have been used.

Results:

In general, avoid re-iterating all the results from the table by text, instead just highlight the principal findings.

Data presentation, for a more manageable data presentation consider simplified the tables and layout.

Characteristics of eligible studies:

In table 2 (Characteristics of participating women in the eligible studies) you include low education level – but have not mentioned it earlier on – and its not used after – so the information doesn’t seem relevant.

In the quality assessment of the studies: You assessed the risk of bias, but not the the overall risk of bias, which is used to know the validity of the results, so it would be good to present it.

Under the effectiveness and safety of exercise during pregnancy: page 11, its mentioned between study heterogeneity again. See earlier comment about heterogeneity

You present odds ratio, Participants analyzed intervention /control and GDM events, and n (%) intervention /control, in table 4 and figure 2, There is no need to present that twice, only present it once.

On page 14, in table 5 (other bias), you have 7 studies that get high risk of bias, but you don’t state why.

Discussion

There is some overlapping between Introduction and Discussion sections.

The reason for heterogeneity is not discuss, even though you earlier have stated this “We performed meta-analysis with special emphasis on issues of potential biases and sources of heterogeneity between studies”. Therefore, it would make sense to analyze and elaborate what causes the heterogeneity.

You write: “A subsequent 16 meta-analysis [18] did not show any benefit of exercise, metformin, vitamin D, and probiotics as compared to placebo or no treatment among high-risk women. These conflicting results indicated that there should be some skepticism about trials on GDM preventive interventions.”

The article: Pascual-Morena C, Cavero-Redondo I, Álvarez-Bueno C, et al. Exercise versus Metformin to Improve Pregnancy Outcomes among Overweight Pregnant Women: A Systematic Review and Network Meta-Analysis. J Clin Med. 2021;10(16):3490. Published 2021 Aug 7. doi:10.3390/jcm10163490. Find that aerobic exercise showed an effect on GDM– which support your finding. – how does that effect the believe in you finding?

You write that future studies should: intensive monitoring to minimize losses to follow up

- but a lot of the events that caused loss is due to change in pregnancy like miscarried/ premature delivery or fetal death in utero, and those events you can’t change.

In the result section you used a lot of space comment on the “Characteristics of the included studies”, but you do not comment on what this means for the results in the article.

6. PLOS authors have the option to publish the peer review history of their article (what does this mean?). If published, this will include your full peer review and any attached files.

Reviewer #1: No

Reviewer #2: No

---

## [Author Response · Author response to Decision Letter 0]

25 Mar 2022

Responses to Editor and Reviewers

Editor’s comments

1) In the literature search was the MeSH terms used or was the search just performed as text words. The use of the filter – randomized controlled trial – may be less sensitive than the Cochrane highly sensitive filter for identifying RCT (these issues need to be addressed as limitation). 

REPLY: We thank the Editor for this comment. We performed again the search in the three databases according to the Editor’s suggestion. We rephrased to “For Pubmed, we used a search strategy including keywords related to exercise, physical activity, and GDM combined with the Cochrane Collaboration search algorithm for RCTs. We conducted a systematic search on Scopus using the same keywords after excluding articles registered in Pubmed. Finally, we searched CENTRAL including the same keywords related to exercise, physical activity, and GDM. Search algorithms were described in detail in Table S1.” [In Material and methods, Search strategy] 

2) The study was not protocolized in either open science framework (www.osf) nor in PROSPERO. This is a severe limitation and need to be addressed too.

REPLY: As suggested, we protocolized the study in the open science framework (www.osf). Specifically, we added that “Our study was pre-registered in the Open Science Framework (OSF) (Registration DOI 10.17605/OSF.IO/23NJS, https://archive.org/details/osf-registrations-23njs-v1).” [In Material and methods, first paragraph]

3) The study was reported according to the PRISMA guidelines – using the PRISMA flowchart may improve the readability and the quality of the flowchart. Further the reporting in the flowchart is inconsistent – 780 hits was retrieved and 72 was duplicates – but only 474 was screened – and 639 were excluded based on title and abstract. Further some difference between text and flowchart are seen (73 or 72 duplicates).

REPLY: We thank the Editor for this comment. After performing the new search strategy, we realized during the screening that several of the RCTs that were included as eligible in the manuscript we initially submitted, evaluated an intervention that included a diet component in addition to exercise. This was in contrast with our initial inclusion criteria. Since those trials were erroneously included in the first manuscript, after our revised search, we excluded them. We have now presented the correct numbers in the text “Our search yielded 1267 items (458 in PubMed, 235 in Scopus, and 574 in CENTRAL). We excluded 216 as duplicated. Out of the 1051 remaining items, we excluded 1019 as non-relevant based on the title, or abstract. Thus, we retrieved 32 papers in full text. Out of the 32 articles, we excluded 25; one paper reported a pilot study; five studies did not include an eligible population; 7 studies included a non-eligible intervention; and 12 trials did not report the onset of GDM as an outcome. Finally, we included 7 published RCTs as eligible for our study (Figure 1)”. In addition, we revised the flow chart (Figure 1).

4) The author used Odds Ratio (OR) for pooling the results – even though OR showed more extreme results compared to Relative Risk (RR) and all included studies are RCT making a meta-analysis on RR possible. Presenting the results in RR may easy the interpretability of the results for clinician. 

REPLY: We agree with the Editor that RR may have easy interpretability. However, odds ratios have favorable mathematical properties; and therefore, we chose to complete our analyses with OR. In addition, the assumed control risk (ACR 0.30) cannot be considered high. Thus, the two estimates (OR and RR) coincide (OR = RR = 0.70). 

5) Further the plots are quite blurry (both the forest plot and the flowchart).

REPLY: As suggested, we revised both the forest plot (Figure 2) and the flowchart (Figure 1) to improve image resolution.

6) The author stated that they have especial focus on explaining the heterogeneity - however why is the only analysis for investigating heterogeneity Eggers test for small study bias – what about the amount of exercise – the duration of the exercise intervention – the risk of bias – et cetera. These analysis can be performed comparing subgroups or investigated in a meta-analysis.

REPLY: As suggested, we added that “We performed separate analyses limited to studies where increased BMI was included in as a risk factor for GDM, and studies that did not consider BMI; studies where the percentage of participating women with low level education was more than 5%; studies that evaluated an intervention delivered individually, and studies that evaluated an intervention delivered in a group; trials that included a motivation component in the intervention, and trials that did not include motivation; studies with an intervention duration more than 20 weeks, and studies with an intervention duration up to 20 weeks. We also performed meta-regression analyses on GDM OR. The effect of baseline risk, and study duration were included individually as covariates in the meta-regressions. For each meta-regression, the slope coefficient with the standard error (SE), the permutation-based P value (as suggested by Higgins and Thompson [27] and the tau2 were reported” [In Material and methods, Statistical analysis]

We also added “The summary odds ratio did not change much when sensitivity analyses were limited to studies fulfilling specific criteria including the use of increased BMI as a risk factor for GDM; the use of a motivation component in the intervention; and the type of intervention delivery (individually, or in a group) (Table S3). However, the summary odds ratio suggested a protective effect for studies with more than 5% of the participating women reporting a low education level (OR 0.49, 95%CI 0.43, 0.73; P-value 0.0006); and for studies that evaluated an intervention with duration more than 20 weeks (OR 0.47, 95%CI 0.31, 0.73; P-value 0.0007). Τhere was no between study heterogeneity for all sensitivity analyses except for one (Table S3). However, we could not exclude large variability in study effects due to real study differences for all sensitivity analysis. (Table S3) Thus, even statistically significant effects should be interpreted with caution because the true differences in effects across studies might be due to unidentified or unexplained underlying factors. Meta-regression analyses with baseline risk, and study duration as covariates did not show a statistically significant effect on the summary OR (Table S4)” [In Results, Effectiveness and safety of exercise during pregnancy] 

Finally, we added two supplementary tables (Table S3 for subgroup and sensitivity analyses; and Table S4 for meta-regression analyses).

7) Finally, the GRADE approach should be used for evaluating the overall evidence (based on study limitation, inconsistency, indirectness, impression and publication bias)

REPLY: We thank the Editor for this comment. As suggested, we added “In addition, we used the Grading of Recommendations, Assessment, Development and Evaluation tool (GRADE) for rating the overall evidence [24] (GRADEpro, Version 3.6.1 McMaster University, 2011)”. [In Material and methods, Quality assessment of the studies and rating of overall evidence]

We also added “Overall, moderate quality of evidence showed that exercise during pregnancy in high-risk women may not have benefit when compared to standard prenatal care in reducing the risk of GDM. The level of evidence for RCTs was downgraded due to the unreported or lack of blinding in participants, personnel, and outcome assessors in most of the studies and because the optimal information size was not met.” [In Results, Quality of reporting, potential bias, and quality of evidence] We also added Table S5.

Reviewer 1

1) The shortcoming of the manuscript is the difficulty of addressing the impact of the findings and viewing the results in the context of similar studies. What is the impact of this meta-analysis compared to the other meta-analysis on the topic?

REPLY: We thank the reviewer for this comment. We added that “However, to our knowledge, there was no systematic approach to evaluate exercise as a single intervention during pregnancy on GDM prevention among high-risk women with any of the risk factors for GDM, and who already received standard prenatal care.” [In Introduction, third paragraph, last sentence]

2) Abstract page 3: The last four lines of the Results section are repeated in the conclusion. Consider omitting the four lines in the results section to include more results from the meta-analysis. The first line of the conclusion should address the aim of the study.

REPLY: As suggested, we omitted the last four lines in the Results section in Abstract. We also rephrased our Conclusion to “Our study did not support a protective effect of exercise intervention during pregnancy for high-risk women to prevent GDM.” [In Abstract, Conclusions]

3) Introduction page 4: The authors mention several risk factors for GDM but in the research question it is not clear how they define “high-risk pregnant women”. Do the women have 1, 2, 3 4 or all of the mentioned risk factors to be at high risk. The definition of “high-risk pregnant women” should be elucidated already in the introduction?

REPLY: As suggested, we clarified in the research question (last paragraph in Introduction) that “We included RCTs on high-risk pregnant women with one or multiple risk factors…”.

4) Methods section (Eligibility Criteria) page 5: Ethnicity as a risk factor could warrant elaboration since it is not as intuitive as the other risk factors.

REPLY: We thank the reviewer for this comment. As suggested, we clarified “…non-white ethnicities…”. [In Methods, Eligibility criteria]

5) Discussion section page 15-16: To strengthen the discussion section, it would be beneficial with a more direct comparison of your study to [ref 18] and [ref 19].

[ref 18]: This meta-analysis found no effect of physical exercise on GDM. Why is their conclusion different to yours? Differences in eligibility criteria for included studies or differences in the characteristics of participating women in the eligible studies?

REPLY: We clarified that “A meta-analysis that evaluated exercise initiated before pregnancy among overweight or obese women showed a reduction in GDM incidence [19]. However, our study focused on RCTs that evaluated an exercise intervention during pregnancy, which might shed light on whether interventions during that period are worth being considered. Based on our subgroup analysis, future research on exercise interventions with adequate duration among pregnant women might be promising. … In line with our study, a subsequent meta-analysis [18] did not show any benefit of exercise, metformin, vitamin D, and probiotics as compared to placebo or no treatment among obese pregnant women. However, our meta-analysis followed a more pragmatic approach for population eligibility including not only studies with pregnant women with increased BMI but also studies with pregnant women with other risk factors for GDM.” [In Discussion, paragraph 4]

6) [ref 19]: This meta-analysis also finds a beneficial effect of exercise but with fewer reservations about their result. Is it only because of prenatal exercise or small sample sizes in included studies? Be kind to elaborate – what does the author's scepticism consist of?

REPLY: As suggested, we clarified “Based on our subgroup analysis, future research on exercise interventions with adequate duration among pregnant women might be promising. However, this result should be interpreted cautiously since a large variability in study effects could not be excluded.” [In Discussion, paragraph 4]

7) Discussion section page 17: The authors have to further elaborate on how this study adds to what is already known on exercise to prevent GDM apart from identifying the presence of publication bias. However, the authors succeed in suggesting improvements in future RCT such as an adequate description of the exercise intervention and motivational techniques for participants to complete the intervention.

REPLY: As suggested, we rephrased “Our findings could not support on average a protective effect of exercise intervention during pregnancy for GDM prevention. Based on our sensitivity analyses, a beneficial effect cannot be excluded among women with low education level, and for exercise interventions with a more than 20-week duration. However, even for these two analyses, potential large true differences in effects among studies could not be excluded. There may be additional unidentified or unexplained underlying factors that may account of the differences in effects. Future large, good quality trials recruiting pregnant women of low education level and evaluating an exercise intervention with satisfactory duration need to adequately report on the intervention characteristics to allow for evaluating potential frequency-response relationship between exercise and GDM risk reduction [12].” [In Discussion, paragraph 6]

Reviewer 2

Abstract:

1) The phrase “between-study heterogeneity was estimated” is used multiple times during the article.

REPLY: As suggested we explained that “Between study heterogeneity (Cochran’s Q statistic) and the extent of study effects variability [I2 with 95% confidence interval (CI)] were estimated.” [In Abstract, Methods section]

We also clarified the phrase “…between study heterogeneity…” throughout the whole manuscript.

2) Ex. page 3: “There was significant between study heterogeneity [Q 24.45, P-value 0.004; I2 = 63% (95%CI 27%, 81%)]”.

It seems like you have mixed two expressions, heterogeneity is between study variance above the expected (chance), so you have to decide to write between study variance or heterogeneity. Besides that, one should separate the values, so one cite the Q and I value for themself. because I is the proportion of the total variation due to inconsistency, and Q is saying if the variation is random (If Q is significant (p < 0.05) => heterogeneity).

REPLY: As suggested, we rephrased to “Τhere was no between study heterogeneity (Q 0.90, P-value 0.64 for low education level; and Q 0.82, P-value 0.84 for duration more that 20 weeks). However, we could not exclude large variability in study effects (I2 0; 95%CI 0, 90%; and I2 0; 95%CI 0, 85% respectively)”. [In Abstract, Results section]

Introduction:

3) In the introduction it would be relevant to mention education levels impact on GDM, since it is mentioned in table 2 later.

Reply: As suggested, we added “Identified risk factors for GDM include…socioeconomic factors including low education level [14]…” [In Introduction, 2nd paragraph]

4) On page 4 it is stated that “We performed meta-analysis with special emphasis on issues of potential biases and sources of heterogeneity between studies.” Here it would be good to analyze what the heterogeneity is, and not only if there is publication bias.

REPLY: As suggested, we rephrased to “We performed meta-analysis with special emphasis on … and sources of study heterogeneity including both clinical and methodological factors that may account for potential variability in study effects.” [In Introduction, last paragraph]

Methods:

5) Some parts of the Methods section are not adequately explained

In the introduction one writes “Identified risk factors for GDM include obesity [1,4-6,9-13], sedentary lifestyle [10], unbalanced diet [3,10], ethnicity [2,4,10,14] and family history [4,10] “and then in the Eligibility Criteria you elaborate, but it is highly recommended to elaborate on ethnicity as well. Besides that, sedentary lifestyle and unbalanced diet is mentioned as a risk factor, but not used as an eligibility Criteria. What are your thoughts about that specific? A small comment about this would be good to include.

REPLY: We thank the reviewer for this comment. As suggested, we clarified “…non-white ethnicities…”. We also rephrased to “Factors that increased pregnant women’s risk included at least one of the following: increased BMI [1,4-6,9-13], sedentary lifestyle [10], family history [4,10,21], previous macrosomia [21], unbalanced diet [3,10], previous GDM [21], non-white ethnicities [2,4,10,14,21] and age > 25 years [21].” [In Methods, Eligibility criteria]

6) In the data extraction on page 6 you mention that you extract level of education, but it’s not used it for anything, please think about why you extract it.

REPLY: We thank the reviewer for this comment. We added that “We also performed separate analyses limited to … studies where the percentage of participating women with low education level was more than 5%.” [In Methods, Statistical analysis]

7) Under the quality assessment of the studies its written that: “We used the risk of bias tool proposed by the Cochrane Collaboration for quality assessment of eligible RCTs “. But it’s not used for anything. Its only too state the findings, but not what it means for the interpretation of the study as a whole.

REPLY: We thank the reviewer for this comment. As suggested, we added that “We performed separate analyses for studies with low detection bias (studies reporting blinding of outcome assessors); and for studies with low attrition bias (studies with less than 20% of participants lost in follow-up).” [In Methods, Statistical analysis]

8) It’s not mentioned how many authors asses the quality of the studies, this is first mentioned during the presentation of the results, consider to mention this in method section instead of in the results. Besides that, you should mention what outcome you asses.

REPLY: As suggested, we added that “Two independent researchers (GIT and KP) extracted the data on quality assessment. Discrepancies were resolved with consensus, and the participation of a third arbitrator (AT) where necessary.” [In Methods, Quality assessment of the studies]

We also added that “Finally, we recorded the number of GDM events as the outcome, separately in the experimental and the control arm. We also captured information on the method used in each study for the diagnosis of GDM.” [In Methods, last paragraph of Data extraction]

9) Under statistical Analysis on page 6 you write that “Heterogeneity was evaluated with Cochran’s Q statistic (statistically significant for P < 0.10)” and later you write “The level of significance was set at P < 0.05”. Hence, it would be good to clarify what significant level that have been used.

REPLY: According to the reviewer, we clarified the “The level of significance for all analyses, except for Cochran’s Q statistic, was set at P-value < 0.05”. [In Methods, Statistical analysis]

Results:

10) In general, avoid re-iterating all the results from the table by text, instead just highlight the principal findings.

REPLY: As suggested, we rephrased our text in Results so that it highlights the principal findings.

11) Data presentation, for a more manageable data presentation consider simplified the tables and layout.

REPLY: As suggested, we tried to simplify the tables and layout to facilitate data presentation.

12) Characteristics of eligible studies:

In table 2 (Characteristics of participating women in the eligible studies) you include low education level – but have not mentioned it earlier on – and it’s not used after – so the information doesn’t seem relevant.

REPLY: We added “We performed separate analyses limited to studies … where the percentage of participating women with low level education was more than 5%...”. [In Methods, Statistical analysis

We also added “…the summary odds ratio suggested a protective effect for studies with more than 5% of the participating women reporting a low education level (OR 0.49, 95%CI 0.43, 0.73 P-value 0.0006);” [In Results, Effectiveness and safety of exercise during pregnancy]

13) In the quality assessment of the studies: You assessed the risk of bias, but not the the overall risk of bias, which is used to know the validity of the results, so it would be good to present it. 

REPLY: As suggested, we rephrased “Based on the overall risk, four trials were judged to raise some concerns because they failed to report specific quality domains. Specifically, three trials [2,13,32] did not provide information on participants and personnel blinding, and on blinding of outcome assessors; and one study [30] did not provide information on participants and personnel blinding only. Three RCTs were judged to be at high risk of bias. One of them [33] did not provide information on participants and personnel blinding, and on blinding of outcome assessors; in addition, it reported a drop-out rate at 31.9%. The other two trials [5,31] were unblinded for participants and personnel; one of them [5] was also unblinded for outcome assessors. (Table 5).” [In Results, Quality of reporting and potential bias] 

14) Under the effectiveness and safety of exercise during pregnancy: page 11, its mentioned between study heterogeneity again. See earlier comment about heterogeneity.

REPLY: As suggested, we rephrased to “When the seven trials were combined, there was no between study heterogeneity (Q 8.34, P-value 0.21). However, we could not exclude large variability (upper limit for Ι2 > 50%) in study effects due to real study differences [I2 28% (95%CI 0%, 69%)]”. [In Results, Effectiveness and safety of exercise during pregnancy]

15) You present odds ratio, Participants analyzed intervention /control and GDM events, and n (%) intervention /control, in table 4 and figure 2, There is no need to present that twice, only present it once.

REPLY: As suggested, we deleted the columns “Participants analyzed, intervention /control”, “GDM events, intervention /control”, and “Odds ratio (95% CI)” in Table 4.

16) On page 14, in table 5 (other bias), you have 7 studies that get high risk of bias, but you don’t state why.

REPLY: We thank the reviewer for this comment. After assessing the quality of the eligible studies, we concluded that indeed there was no high risk for the “Other bias” domain according to the Cochrane guidance on how to address the specific domain. In addition, we have changed the indication to “L” in the column “Other bias” in Table 5.

Discussion

17) There is some overlapping between Introduction and Discussion sections.

REPLY: As suggested, we rephrased the Discussion section to avoid overlapping with Introduction.

18) The reason for heterogeneity is not discuss, even though you earlier have stated this “We performed meta-analysis with special emphasis on issues of potential biases and sources of heterogeneity between studies”. Therefore, it would make sense to analyze and elaborate what causes the heterogeneity.

REPLY: We thank the reviewer for this comment. We added “Notably all trials except for one suggested no benefit, and a small study effect was unlikely. However, we noticed that a large variability in study effects could not be excluded. Several subsequent subgroup and sensitivity analyses did not identify a clinical or methodological factor that may explain for a potential large variability.” [In Discussion, paragraph 1]

19) You write: “A subsequent 16 meta-analysis [18] did not show any benefit of exercise, metformin, vitamin D, and probiotics as compared to placebo or no treatment among high-risk women. These conflicting results indicated that there should be some skepticism about trials on GDM preventive interventions.”

REPLY: We thank the reviewer for this comment. We rephrased to “In line with our study, a subsequent meta-analysis [18] did not show any benefit of exercise, metformin, vitamin D, and probiotics as compared to placebo or no treatment among obese pregnant women. However, our meta-analysis followed a more pragmatic approach for population eligibility including not only studies with pregnant women with increased BMI but also studies with pregnant women with other risk factors for GDM.” [In Discussion, paragraph 4]

20) The article: Pascual-Morena C, Cavero-Redondo I, Álvarez-Bueno C, et al. Exercise versus Metformin to Improve Pregnancy Outcomes among Overweight Pregnant Women: A Systematic Review and Network Meta-Analysis. J Clin Med. 2021;10(16):3490. Published 2021 Aug 7. doi:10.3390/jcm10163490. Find that aerobic exercise showed an effect on GDM– which support your finding. – how does that effect the believe in you finding?

REPLY: We thank the reviewer for this comment. We added “Another meta-analysis [34] that evaluated the effect of different types of exercise and metformin for pregnancy outcomes in overweight and obese pregnant women, showed a reduced risk for GDM with aerobic exercise. However, our meta-analysis followed a more pragmatic approach for population eligibility including not only studies with pregnant women with increased BMI but also studies with pregnant women with other risk factors for GDM.” [In Discussion, paragraph 4]

21) You write that future studies should: intensive monitoring to minimize losses to follow up-but a lot of the events that caused loss is due to change in pregnancy like miscarried/ premature delivery or fetal death in utero, and those events you can’t change.

REPLY: To clarify, we rephrased to “…intensive monitoring to minimize losses to follow up that are not due to miscarriage, premature delivery, or fetal death in utero…”. [In Discussion, paragraph 6]

22) In the result section you used a lot of space comment on the “Characteristics of the included studies”, but you do not comment on what this means for the results in the article.

REPLY: However, another explanation might be that exercise may be effective in specific settings, for specific population subgroups, or when specific intervention characteristics exist. Our study did not show that baseline risk for GDM, using increased BMI as a risk factor or a small percentage of women with low education level in the population might change the effect. Furthermore, including a motivation component in the intervention, the mode of intervention delivery or an intervention duration up to 20 weeks did not change the summary estimate. Based on our subgroup analyses however, a protective effect may be suggested for exercise among women with low education level, and when the intervention duration is more than 20 weeks.” [In Discussion, paragraph 2]

---

## [Decision Letter · Decision Letter 1]

29 Apr 2022

PONE-D-21-21724R1Effectiveness of exercise intervention during pregnancy on high-risk women for gestational diabetes mellitus prevention: a meta-analysis of published RCTsPLOS ONE

Dear Dr. Tatsioni,

Thank you for submitting your manuscript to PLOS ONE. After careful consideration, we feel that it has merit but does not fully meet PLOS ONE’s publication criteria as it currently stands. Therefore, we invite you to submit a revised version of the manuscript that addresses the points raised during the review process.

We look forward to receiving your revised manuscript.

Kind regards,

Carsten Bogh Juhl, PhD

Academic Editor

PLOS ONE

Additional Editor Comments (if provided):

Even though the authors have addressed some of the comments from the editor and the reviewer this manuscript is still insufficient for publication. The author has added the following to the abstract – however the sentence does not really make sense - However, we could not exclude large variability in study effects (I2 0; 95%CI 0, 90%; and I2 0; 95%CI 0, 85% respectively). The author stated that the study was preregistered even though the registration was performed after the first submission.

The quality of the forest plot, funnel plot and the flowchart is still too low for printing. The figures are not sufficiently self-explanatory – most of the figures need more explanation. Figure 1 presenting the search is still insufficient (does not define Mesh and text words for searching in Medline and Central – even though the author stated that the search was updated – but the one added in table one is limited up-to July 2020 and in Scopus studies from 2021 and 2022 is deleted. Table 3 are showing a subgroup analysis – however presenting the Q, I-square and p-value for the subgroup – however the test of difference may be the one of clinical interest. Table 4 is not clear and need more explanation. Table 5, the SOF table need explanation for what is study population and what does moderate means – the evidence is judged moderate even though the confidence interval is very broad and overlapping 1 (indicating no significant effect).

Reviewers' comments:

Reviewer's Responses to Questions

**Comments to the Author**

1. If the authors have adequately addressed your comments raised in a previous round of review and you feel that this manuscript is now acceptable for publication, you may indicate that here to bypass the “Comments to the Author” section, enter your conflict of interest statement in the “Confidential to Editor” section, and submit your "Accept" recommendation.

Reviewer #1: All comments have been addressed

Reviewer #2: (No Response)

2. Is the manuscript technically sound, and do the data support the conclusions?

Reviewer #1: (No Response)

Reviewer #2: Yes

3. Has the statistical analysis been performed appropriately and rigorously? 

Reviewer #1: (No Response)

Reviewer #2: N/A

4. Have the authors made all data underlying the findings in their manuscript fully available?

Reviewer #1: (No Response)

Reviewer #2: Yes

5. Is the manuscript presented in an intelligible fashion and written in standard English?

Reviewer #1: (No Response)

Reviewer #2: Yes

6. Review Comments to the Author

Reviewer #1: (No Response)

Reviewer #2: The protocol ismade after the first submission, which is a big limitation. The search is very deficient, and lacks Mesh terms and many relevant keywords, resulting in missing relevant articles (Unless table 1, isn't updated since last submission)

The following 3 articles would be relevant to include, and meet your inclusion criteria:

1. Barakat R, Pelaez M, Lopez C, Lucia A, Ruiz JR. Exercise during pregnancy and gestational diabetes-related adverse effects: a randomized controlled trial. Br J Sports Med. 2013 Jul; 47 (10): 630-6. doi: 10.1136 / bjsports-2012-091788. Epub 2013 Jan 30. PMID: 23365418.

(Inclusion criteria included: being sedentary, with match yours)

2. Seneviratne SN, Jiang Y, Derraik J, McCowan L, Parry GK, Biggs JB, Craigie S, Gusso S, Peres G, Rodrigues RO, Ekeroma A, Cutfield WS, Hofman PL. Effects of antenatal exercise in overweight and obese pregnant women on maternal and perinatal outcomes: a randomized controlled trial. BJOG. 2016 Mar; 123 (4): 588-97. doi: 10.1111 / 1471-0528.13738. Epub 2015 Nov 6. PMID: 26542419.

(Participants were women aged 18–40 years with a body

mass index (BMI) ≥25 kg / m2,)

3. From Oliveria Melo AS, Silva JL, Tavares JS, Barros VO, Leite DF, Amorim MM. Effect of a physical exercise program during pregnancy on uteroplacental and fetal blood flow and fetal growth: a randomized controlled trial. Obstet Gynecol. 2012 Aug; 120 (2 Pt 1): 302-10. doi: 10.1097 / AOG.0b013e31825de592. PMID: 22825089.

(The inclusion criteria consisted of: healthy pregnant women who were sedentary at admission to the

study)

Many of the previous fixes have been improved, but the search should have been significantly improved.

7. PLOS authors have the option to publish the peer review history of their article (what does this mean?). If published, this will include your full peer review and any attached files.

Reviewer #1: **Yes: **Jørgen Guldberg-Møller

Reviewer #2: No

---

## [Author Response · Author response to Decision Letter 1]

12 Jun 2022

Responses to Editor and Reviewers

Editor’s comments

1) The author has added the following to the abstract – however the sentence does not really make sense - However, we could not exclude large variability in study effects (I2 0; 95%CI 0, 90%; and I2 0; 95%CI 0, 85% respectively). 

REPLY: We thank the editor for this comment. We rephrased to “However, we could not exclude large variability in study effects because the upper limit of I² confidence interval was higher than 50% for all the analyses.”

2) The author stated that the study was preregistered even though the registration was performed after the first submission.

REPLY: We rephrased to “registered”. (Page 6; Paragraph 1)

3) The quality of the forest plot, funnel plot and the flowchart is still too low for printing.

REPLY: As suggested, we improved the quality of the forest plot, funnel plot and the flowchart.

4) The figures are not sufficiently self-explanatory – most of the figures need more explanation.

REPLY: We thank the Editor for this comment. We have now rephrased the Legends for the three figures to sufficiently explain their content. 

5) Figure 1 presenting the search is still insufficient (does not define Mesh and text words for searching in Medline and Central – even though the author stated that the search was updated – but the one added in table one is limited up-to July 2020 and in Scopus studies from 2021 and 2022 is deleted. 

REPLY: As suggested, we updated our search up to May 2022. Table S1 includes the exact MeSH terms and keywords, we used. We have also updated Figure 1. The number of articles in the Flowchart (Figure 1) correspond to the numbers we found in our last search (May 2022).

6) Table 3 are showing a subgroup analysis – however presenting the Q, I-square and p-value for the subgroup – however the test of difference may be the one of clinical interest.

REPLY: As suggested, we added the P-value for the test of difference for all subgroup analyses in the revised Table S3. 

7) Table 4 is not clear and need more explanation.

REPLY: As suggested, we clarified Table 4, and provided additional explanation “GDM was diagnosed by measuring fasting blood glucose, hemoglobin A1c, or by an oral glucose tolerance test (Table 4).” (Page 14; paragraph 2)

8) Table 5, the SOF table need explanation for what is study population and what does moderate means – the evidence is judged moderate even though the confidence interval is very broad and overlapping 1 (indicating no significant effect).

REPLY: As suggested we clarified “Five out of the 9 studies had potential performance, detection, or attrition bias. The other 4 studies were unclear about blinding. Overall, moderate quality of evidence showed that exercise during pregnancy for the population of women with high risk for GDM may have benefit when compared to standard prenatal care in reducing the risk of GDM. (Table S5)” (Page 17, Paragraph 1) 

Reviewer 1

All comments have been addressed.

REPLY: We thank the Reviewer for this comment.

Reviewer 2

1) The protocol is made after the first submission, which is a big limitation. The search is very deficient, and lacks Mesh terms and many relevant keywords, resulting in missing relevant articles (Unless table 1, isn't updated since last submission)

REPLY: We thank the Reviewer for this issue. As suggested, we have updated our initial search strategy including additional MeSH terms and keywords and expanding our search (from inception) to May 2022. We have updated Table S1 to include the exact search strategy. We have also updated all Tables, and Figures in the Results so that they include the additional studies. The text has also been updated as appropriate.

2) The following 3 articles would be relevant to include, and meet your inclusion criteria:

1. Barakat R, Pelaez M, Lopez C, Lucia A, Ruiz JR. Exercise during pregnancy and gestational diabetes-related adverse effects: a randomized controlled trial. Br J Sports Med. 2013 Jul; 47 (10): 630-6. doi: 10.1136 / bjsports-2012-091788. Epub 2013 Jan 30. PMID: 23365418.

(Inclusion criteria included: being sedentary, with match yours)

2. Seneviratne SN, Jiang Y, Derraik J, McCowan L, Parry GK, Biggs JB, Craigie S, Gusso S, Peres G, Rodrigues RO, Ekeroma A, Cutfield WS, Hofman PL. Effects of antenatal exercise in overweight and obese pregnant women on maternal and perinatal outcomes: a randomized controlled trial. BJOG. 2016 Mar; 123 (4): 588-97. doi: 10.1111 / 1471-0528.13738. Epub 2015 Nov 6. PMID: 26542419.

(Participants were women aged 18–40 years with a body mass index (BMI) ≥25 kg / m2,)

3. From Oliveria Melo AS, Silva JL, Tavares JS, Barros VO, Leite DF, Amorim MM. Effect of a physical exercise program during pregnancy on uteroplacental and fetal blood flow and fetal growth: a randomized controlled trial. Obstet Gynecol. 2012 Aug; 120 (2 Pt 1): 302-10. doi: 10.1097 / AOG.0b013e31825de592. PMID: 22825089.

(The inclusion criteria consisted of: healthy pregnant women who were sedentary at admission to the

study)

REPLY: Indeed, we have identified the three papers the Reviewer has indicated. As suggested, we have now added Bakarat et al. 2013, and Seneviratne et al. 2016 in our updated search and analyses. However, we had to exclude Oliveiria Melo et al. 2012 because it did not fulfill our inclusion criteria. Specifically, it did not report the diagnosis of GDM as one of its outcomes.

---

## [Decision Letter · Decision Letter 2]

26 Jul 2022

Effectiveness of exercise intervention during pregnancy on high-risk women for gestational diabetes mellitus prevention: a meta-analysis of published RCTs

PONE-D-21-21724R2

Dear Dr. Tatsioni,

We’re pleased to inform you that your manuscript has been judged scientifically suitable for publication and will be formally accepted for publication once it meets all outstanding technical requirements.

Kind regards,

Carsten Bogh Juhl, PhD

Academic Editor

PLOS ONE

Additional Editor Comments (optional):

Reviewers' comments:

Reviewer's Responses to Questions

**Comments to the Author**

1. If the authors have adequately addressed your comments raised in a previous round of review and you feel that this manuscript is now acceptable for publication, you may indicate that here to bypass the “Comments to the Author” section, enter your conflict of interest statement in the “Confidential to Editor” section, and submit your "Accept" recommendation.

Reviewer #2: All comments have been addressed

2. Is the manuscript technically sound, and do the data support the conclusions?

Reviewer #2: Yes

3. Has the statistical analysis been performed appropriately and rigorously? 

Reviewer #2: Yes

4. Have the authors made all data underlying the findings in their manuscript fully available?

Reviewer #2: Yes

5. Is the manuscript presented in an intelligible fashion and written in standard English?

Reviewer #2: Yes

6. Review Comments to the Author

Reviewer #2: (No Response)

7. PLOS authors have the option to publish the peer review history of their article (what does this mean?). If published, this will include your full peer review and any attached files.

Reviewer #2: No

---

## [Editor Report · Acceptance letter]

28 Jul 2022

PONE-D-21-21724R2 

Εffectiveness of exercise intervention during pregnancy on high-risk women for gestational diabetes mellitus prevention: a meta-analysis of published RCTs 

Dear Dr. Tatsioni:

I'm pleased to inform you that your manuscript has been deemed suitable for publication in PLOS ONE. Congratulations! Your manuscript is now with our production department. 

Kind regards, 

on behalf of

Dr. Carsten Bogh Juhl 

Academic Editor

PLOS ONE